# 3D Global Localization in the Underground Mine Environment Using Mobile LiDAR Mapping and Point Cloud Registration

**DOI:** 10.3390/s22082873

**Published:** 2022-04-08

**Authors:** Jieun Baek, Junhyeok Park, Seongjun Cho, Changwon Lee

**Affiliations:** Mineral Resources Division, Korea Institute of Geoscience and Mineral Resources, Daejeon 34132, Korea; bje0511@gmail.com (J.B.); junhpark@kigam.re.kr (J.P.); mac@kigam.re.kr (S.C.)

**Keywords:** global localization, LiDAR, point cloud registration, digital twin, underground mine

## Abstract

This study proposes a 3D global localization method that implements mobile LiDAR mapping and point cloud registration to recognize the locations of objects in an underground mine. An initial global point cloud map was built for an entire underground mine area using mobile LiDAR; a local LiDAR scan (local point cloud) was generated at the point where underground positioning was required. We calculated fast point feature histogram (FPFH) descriptors for the global and local point clouds to extract point features. The match areas between the global and the local point clouds were searched and aligned using random sample consensus (RANSAC) and iterative closest point (ICP) registration. The object’s location on the global coordinate system was measured using the LiDAR sensor trajectory. Field experiments were performed at the Gwan-in underground mine using three mobile LiDAR systems. The local point cloud dataset formed for the six areas of the underground mine precisely matched the global point cloud, with a low average error of approximately 0.13 m, regardless of the type of mobile LiDAR system used. In addition, the LiDAR senor trajectory was aligned on the global coordinate system to confirm the change in the dynamic object’s position over time.

## 1. Introduction

Recently, digital twin technology, which has emerged as a representative industry 4.0 technology, has been introduced in the mining industry to increase mining productivity, efficiency, and safety [1,2,3]. Digital twins are digital replications of physical objects, models, and systems into the virtual world [4]. In particular, digital twins simultaneously link physical and digital models; they enable real-time monitoring, analysis, prediction, optimization, and automation of the physical model using artificial intelligence, internet of things, cloud, big data, and mobile (i.e., AICBM) technology [5]. Digital twins enable decisions for optimal mine operation by monitoring, controlling, designing, and evaluating the static and dynamic assets (e.g., ores, equipment, workers, and facilities), mining processes (e.g., explosions, hauling, and processing), and environment (e.g., geology and exploration) [6].

To increase the replication precision and prediction accuracy of digital twins, it is important to recognize and measure the 3D spatial coordinates required for accurate digital transformation of static and dynamic assets, work processes, and environmental data. When errors exist in the spatial positioning of digital data, the prediction uncertainty of the digital twins may increase. For example, inaccurate spatial coordinates of 3D geological survey data (e.g., rock type, ore grade, and borehole data) can lead to erroneous ore reserve estimation and mine layout design. In addition, incorrect position coordinates of a dynamic object may degrade the prediction accuracy of mining operations. Furthermore, a workplace accident of the automated path guidance system may occur because of incorrect location recognition and destination setting.

However, unlike open-pit mines, underground mines are global positioning system (GPS)-denied areas; it is thus difficult to recognize, measure, or track accurate location coordinates within underground mines [7]. To overcome this problem, three approaches can be used to recognize locations in underground mines. The first approach involves the use of a total station instrument. The total station begins designating coordinates from the mine portal where GPS functions. It uses laser and artificial landmarks to designate coordinates sequentially from close areas (near the portal) to deep areas (distant from the portal). However, the use of the total station in underground mines has a few limitations. First, the total station calculates the new coordinates by only measuring the distance and vertical/horizontal angles from the known location. This increases the localization uncertainty because the cumulative error increases as the drift deepens. Second, installing a large number of artificial landmarks at dense intervals can be a good option to make autonomous driving equipment continuously recognize 3D coordinates. However, it is costly and time-consuming. Third, frequent drift blasting and excavation causes artificial landmarks to be lost, relocated, or reinstalled.

The second approach involves the use of wireless communication technologies. A wireless signal is transmitted by a sensor attached to an underground mine wall. To identify the location coordinates, the signal is recognized by a receiver mounted on the equipment. Until recently, localization methods used in underground mines were developed mainly by using Wi-Fi [8,9,10,11], ZigBee [12,13,14,15,16], radio frequency identification [17,18,19,20,21,22], and Bluetooth [23,24,25]. However, to enable continuous location recognition and improve recognition accuracy, sensors must be attached to the wall in a concentrated manner, which may lead to high installation costs. In addition, the excavations with crossings inclined at 90 degrees may generate diffraction, reflection, and refraction of wireless signals.

The third approach involves an indoor localization method that employs light detection and ranging (LiDAR)-based simultaneous localization and mapping (SLAM) technology. The LiDAR estimates the distance by converting the travel time required for the laser signal in the infrared frequency band to strike the object and return [26]. LiDAR SLAM technology recognizes the robot’s location while simultaneously generating a point cloud map [27]. Recently, a few studies have been conducted with the goal of recognizing location using LiDAR SLAM in an underground mine. Jacobson [28] developed a new multisensor-based 2D SLAM technology for automatic vehicle operation in an underground mine, while Losch et al. [29] developed a new 3D LiDAR SLAM platform for the underground localization of automatic driving robots.

To improve the accuracy of LiDAR mapping and pose estimation, it is preferable to use auxiliary sensors such as a vision camera and inertial measurement unit. However, underground mines have a high magnetic field intensity and dark working spaces, which prevent the use of auxiliary sensors. Thus, LiDAR odometry and mapping (LOAM) [30] was developed to recognize a robot’s location without auxiliary sensors through edge/plane feature extraction and consecutive 3D scan matching. In addition, lightweight and ground-optimized (LEGO) LOAM [31] has been suggested to complement the scan matching accuracy and cumulative drift error of LOAM via loop detection. A few researchers have proposed an improved LOAM or LEGO-LOAM for real-time localization in underground mines with high precision and low cost. Ren et al. [32] combined LOAM and a 3D generalized-ICP (GICP) algorithm for accurate loop detection and optimization with limited computing resources for an underground mine. Xue et al. [33] suggested an improved LEGO-LOAM fused with scan context for real-time localization in an underground coal mine. Ren and Wang [34] improved the 3D GICP-SLAM to implement real-time localization by introducing an inverse distance weighting (IDW) into the error correction.

There are some limitations to the application of existing LiDAR SLAMs for measurement of the 3D spatial coordinates of the assets, objects, and equipment required for data transformation to the digital twin. The SLAM simultaneously builds a point cloud map and estimates each object’s location in a local coordinate system. The LiDAR begins mapping at the origin of the local coordinate system. Meanwhile, the digital twin platform requires data on a global coordinate system; an object’s location coordinate must be transformed into the global coordinate system using a transformation matrix. However, the transformation matrix is variable because the origin of the local coordinate system is initialized when SLAM builds a new map. Hence, the local point cloud map should be compared with a pre-built global point cloud map of the entire mine area to identify a match area between maps by extracting geometric features of points. Then, the transformation matrix can be obtained by calculating the fine alignment between the local and global point cloud maps using the point cloud registration technique. This process is known as global localization. Global localization estimates the location in the global coordinate system by matching and aligning the pre-built point cloud map and the local LiDAR scan via point feature extraction and point cloud registration [35]. Thus far, studies have been conducted to estimate the location of a self-driving vehicle through 3D map matching between a pre-defined point cloud map and a local LiDAR scan in urban road [36] and outdoor environments [37]. However, no studies have applied LiDAR-based 3D global localization to recognize an object’s location in an underground mine.

The purpose of this study was to present a 3D global localization method that used mobile LiDAR mapping and point cloud registration for the recognition of an object’s location in an underground mine. This method searches the match area between the pre-built point cloud map of the underground mine and the local point cloud through random sample consensus (RANSAC) and iterative closest point (ICP) registration. In addition, it estimates the object’s location in the global coordinate system. We present the results of field experiments using three mobile LiDAR systems in the Gwan-in underground mine in South Korea.

## 2. Methods

The global localization method for an underground mine follows five steps. Figure 1 is a schematic diagram that describes the global localization procedure. First, mobile LiDAR is used to generate a global cloud map of the entire underground mine and transform the coordinates to the Universal Transverse Mercator (UTM) coordinate system in meters. A local point cloud with a range of ≥10 m is generated using the mobile LiDAR in an area that requires location recognition. Second, to increase the point cloud processing efficiency, the resolution of the global point cloud and local point cloud is reduced through octree-based subsampling. Third, through feature-based coarse registration, the matching area between the global and the local point clouds is searched and the transformation matrix is calculated. In this step, the correspondences between the global and local point clouds are iteratively searched using the RANSAC technique. Fourth, the alignment between the global and local point clouds is precisely matched using the ICP algorithm. Fifth, the position coordinates of the object are estimated from the LiDAR sensor trajectory.

### 2.1. Global and Local Point Clouds Generation

For global localization, the formation of a global point cloud for the entire area of an underground mine and the establishment of a global coordinate system must be performed. In this study, the global point cloud is assumed to be produced in advance using mobile LiDAR. In general, point cloud generation using LiDAR follows the scan matching method. Scan matching is a technique used to calculate the relative transformation between correspondences in which geometric features match each other in a pair of overlapping point clouds [38]. As the LiDAR sensor moves, it forms point cloud sets for continuous scenes. It explores the correspondence points in a pair of point clouds and matches their alignment.

The pose and location of the LiDAR sensor can be estimated by calculating the relative transformation between correspondence points. Equation (1) is the equation used for calculating the LiDAR sensor location [30]. Here, *L_τ_* denotes the three-dimensional position coordinates of the LiDAR sensor at time *τ*; Δ*T_τ_*_−1_*,_τ_* denotes the relative transformation according to the time change (from *τ* −1 to *τ*). The relative transformation consists of a rotation matrix and a translation matrix. Figure 2 is a schematic that explains the principle of forming a point cloud map through continuous scan matching and estimation of the LiDAR sensor trajectory.
(1)$$L_{\tau }=L_{\tau -1}\Delta T_{\tau -1,\tau }$$

The initial global point cloud is set as the local frame, in which the coordinates of the starting point of LiDAR scanning are assigned as the origin (0,0,0). Conversion of the initial global point cloud into the UTM coordinate system requires ground control points with known real-world coordinates. The initial global point cloud map is transformed into the UTM coordinate system by calculating the transformation matrix between ground control points and reference points.

The user operates the mobile LiDAR at a point where location coordinates in the mine are required. The local point cloud is generated to have a distance of ≥10 m around the object. Because there is no ground control point information in the local point cloud, automatic conversion to the global coordinate system is impossible. Therefore, global localization can be achieved through point feature extraction and point cloud registration, which are discussed in the next sections.

### 2.2. Point Cloud Down-Sampling

To increase the operating speed and efficiency of the global localization, the resolution of the global and local point clouds is lowered. For down-sampling, an octree structure-based subdivision method is applied [39]. The octree method is one of the methods used for indexing 3D point cloud data. The octree node exists in the form of a rectangular cuboid and is subdivided into eight children. No further subdivision is necessary when the octree node is not occupied by points. Figure 3 describes the subdivision method based on the octree structure. The octree level represents the number of subdivisions. Octree level 1 constitutes the division of the entire point cloud data area by eight octree nodes, while octree level 2 constitutes the division of the occupied octree nodes (gray volumetric cubic area in Figure 3) by the eight children. The octree structure is suitable for effective storage and retrieval of point cloud data that are not fully volumetric.

Global and local point clouds are subdivided by setting the appropriate octree level; only points closest to the center of the octree voxels are sampled. Increasing the octree level reduces the size of the octree cell, allowing more accurate maintenance of drift geometry characteristics. However, because of the increased number of points, considerable time is needed to calculate geometric features. Therefore, the appropriate octree level must be selected after sufficient adjustments.

### 2.3. Match Area Search between Global and Local Point Clouds

In this step, the match area between the global and local point clouds is searched, and they are matched with each other. The wall surface of the underground mine is irregular and inconsistent because of the blasted faces, rough shapes, joint structures, cavities, and ground supports generated during mine excavation. In addition, the drift geometry is not constant locally because of the orebody shape (e.g., scale, slope, and bedding direction) and geological engineering constraints (e.g., joints, faults, and safety ratio). If the irregular geometric and structural characteristics of the underground mine drift are considered, the matching efficiency, accuracy, and quality between the global and local point clouds can be improved. Therefore, in this study, the global and local point clouds were matched with each other by calculation and comparison of the surface normal features of the point clouds.

#### 2.3.1. Surface Feature Extraction Using the FPFH Descriptor

The fast point feature histogram (FPFH) is calculated to visualize the geometrical properties of the global and local point clouds. FPFH is a local feature descriptor proposed by Rusu et al. [40] that expresses surface normal variations between k-nearest neighbors in histograms. The three-step procedure for FPFH calculation is as follows. The following processes apply equally to the global point cloud (*Q_init_*).Surface normal estimation: A down-sampled local point cloud ($P_{init}^{D}$) is generated through subdivision of the local point cloud (*P_init_*) by voxel grids with a specific voxel size. The k-nearest neighbor points enclosed in a sphere with a specific radius (*r*) are extracted for a point (*p*) in the local point cloud ($P_{init}^{D}$). Using principal component analysis, the best-fitting surface including the point *p* and the k-nearest neighbors is extracted. Then, the surface normal for point *p* is calculated by estimation of the smallest eigenvector through singular value decomposition (SVD). Equation (2) is used to calculate the surface normal (*n_i_*) of an *i*th point (*p_i_*), where *V*_0_ is the eigenvector that corresponds to the smallest eigenvector (*λ*_0_). The surface normal is iteratively computed for all *N* points in the down-sampled local point cloud (computational complexity: O(*N*)).
(2)$$n_{i}=V_{0},\lambda_{0}\leq \lambda_{1}\leq \lambda_{2}$$Computation of angular variations in the surface normal: Angular variations in the surface normal are calculated for the k-nearest neighbors of point *p*. The point pi and its nearest neighbor *p_j_* are defined in the Darboux uνw frame *(u = n_i_, v = (p_i_* − *p_j_) × u*, *w = u × v*). Figure 4a depicts the points *p_i_* and *p_j_* in the Darboux frame. The angular variation (*α*, *ϕ*, *θ*) for all nearest neighbors is calculated as shown in Equations (3)–(5). The computational complexity is O(*k*).
(3)$$\alpha =v\times n_{j}$$
(4)$$\phi =u\times \frac{p_{j}-p_{i}}{\left\|p_{j}-p_{i}\right\|}$$
(5)$$\theta =\arctan(w\times n_{j},u\times n_{j})$$Calculation of the FPFH: The angular variation (*α*, *ϕ*, *θ*) is binned into a multi-value histogram known as the simplified point feature histogram (SPFH) via the d^3^ binning scheme. Here, d is the number of subdivision intervals in the feature value range. FPFH is calculated as the sum of the SPFH of *p_i_* and the weighted SPFH of the k-nearest neighbor, as shown in Equation (6). Here, *ω_i_* is the Euclidean distance between *p_j_* and the *j*th k-nearest neighbor. Figure 4b shows the effect area for the FPFH calculation of *p_i_*. The computational complexity of FPFH is O(*N*×*k*).
(6)$$\mathrm{FPFH}(p_{i})=\mathrm{SPFH}(p_{i})+\frac{1}{k}\sum_{j=1}^{k}\frac{1}{\varpi_{j}}\cdot \mathrm{SPFH}(p_{j})$$

Figure 5 shows the FPFH descriptors with 33-dimensional bins calculated for any three points (point 1: mouth, point 2: body, point 3: foot) of the dragon point cloud dataset and the average FPFH descriptor calculated for all points. In Figure 5b, the *x*-axis shows that all SPFH values calculated for point 1 and its k-nearest neighbors are expressed as 33 bins. The *y*-axis represents the ratio of the number of points included in one bin. The FPFH descriptor is calculated according to the geometric characteristics of the point.

#### 2.3.2. Match Area Search Using RANSAC

The match area between global and local point clouds is searched using the FPFH descriptor. Then, the transformation matrix is calculated. The RANSAC technique is used to match the two point clouds. The RANSAC is a model fitting algorithm that estimates the optimal solution while removing outliers via repeated model prediction and verification from experimental data [41]. The RANSAC is used effectively for 3D point cloud registration because it can gradually update the relative transformation that minimizes the distance between the source point cloud and the target point cloud while removing any outliers of the corresponding points [42]. In this study, the global and local point clouds were matched using the modified RANSAC [43] with pruning steps. The process was as follows.Corresponding point set extraction: The down-sampled local point cloud ($P_{init}^{D}$), down-sampled global point cloud ($Q_{init}^{D}$), and their FPFH descriptors are input into the algorithm. Three or more sample points, *p_i_* {*p*_1_, *p*_2_, *p*_3_, …, *p_n_*}, are randomly extracted from the down-sampled local point cloud. The corresponding points, *q_i_* {*q*_1_, *q*_2_, *q*_3_, …, *q_n_*}, of *p_i_* are extracted by querying the nearest neighbors in the d-dimensional FPFH descriptor.Pruning steps: In these steps, the calculation complexity of RANSAC is reduced by extracting only the inlier points via two steps of pruning. First, the relative dissimilarity vector between corresponding points (*p_i_*, *q_i_*) is calculated. The relative dissimilarity vector constitutes the ratio of the difference between the two corresponding edge lengths to the larger edge length calculated from the *n* polygon of corresponding points (*p_i_*, *q_i_*). Figure 6 is a schematic diagram that explains the relative dissimilarity vector computation. Equation (7) is the calculation of the relative dissimilarity vector ($\vec{\eta }$). Here, $d_{1,n}^{p}$ denotes the edge length between *p*_1_ and *p_n_*, while $d_{1,n}^{q}$ denotes the edge length between *q*_1_ and *q_n_*. If the relative dissimilarity vector is greater than the vector threshold (*ε_vector_*), the process returns to step 1.
(7)$$\vec{\eta }=\left[\frac{\left|d_{12}^{p}-d_{12}^{q}\right|}{\max(d_{12}^{p},d_{12}^{q})},\frac{\left|d_{23}^{p}-d_{23}^{q}\right|}{\max(d_{23}^{p},d_{23}^{q})},\ldots \frac{\left|d_{n-1,n}^{p}-d_{n-1,n}^{q}\right|}{\max(d_{n-1,n}^{p},d_{n-1,n}^{q})},\frac{\left|d_{1,n}^{p}-d_{1,n}^{q}\right|}{\max(d_{1,n}^{p},d_{1,n}^{q})}\right]$$Transformation matrix update using RANSAC: A temporary transformation matrix between the correspondence points (*p_i_*, *q_i_*) is calculated, and the down-sampled local point cloud ($P_{init}^{D^{\prime }}$) is transformed to the global point cloud ($Q_{init}^{D}$). Euclidean distances of the nearest neighbors between $P_{init}^{D^{\prime }}$ and $Q_{init}^{D}$ are calculated; only *k* point pairs are extracted as inlier points (*p_k_*, *q_k_*) whose Euclidean distance is lower than the Euclidean distance threshold (*ε_distance_*). Equation (8) is the calculation of a required iteration number *N* of RANSAC; *p* denotes the desired success probability, *w* denotes an expected inlier fraction, and *n* constitutes the number of sample points. Model fitting and inlier point update continue until an optimal transformation matrix that minimizes the sum of the squared distances between the inlier points is obtained or the maximum iteration number *N*_max_ is reached. The initial local point cloud moves to the match area (*P_init_*→*P_RANSAC_*) by transforming the coordinates using the rotation matrix (*R_RANSAC_*) and the movement matrix (*t_RANSAC_*) (Equation (9)).
(8)$$N=\frac{\log\left(1-p\right)}{\log\left(1-w^{n}\right)}$$
(9)$$P_{RANSAC}=R_{RANSAC}P_{init}+t_{RANSAC}$$

### 2.4. Alignment Matching between Global and Local Point Clouds Using ICP

In this step, the local point cloud is precisely aligned on the global point cloud through the ICP algorithm. The ICP algorithm is used to find the optimal rotation matrix and translation matrix that match the alignment of the two point cloud data.Corresponding point set extraction: A random sample point set, *p_i_* {*p*_1_, *p*_2_, *p*_3_, …, *p_n_*}, is extracted in the local point cloud (*P_RANSAC_*), and the correspondence point set, *q**_i_* {*q*_1_, *q*_2_, *q*_3_, …, *q_n_*}, is extracted from the global point cloud (*Q_init_*).Rotation and translation matrix estimation using SVD: Through SVD, a rotation matrix and a translation matrix are calculated for moving the sample point set *p* to *q*. First, the cross-covariance matrix (*W*) for n correspondence point pairs (*p_i_, q_i_*) is calculated by Equation (10). Here, *q_i_^T^* constitutes the transpose matrix of *q_i_*. Next, an *m × n* cross-covariance matrix *W* is decomposed into an *m × m* orthogonal matrix *U*, an *m × n* diagonal matrix Σ, and an *m × n* orthogonal matrix *V* through SVD. Equation (11) is an expression for decomposing the cross-covariance matrix *W* through SVD, where *V^T^* is the transpose matrix of the orthogonal matrix *V*. Through Equations (12) and (13), the rotation matrix *R_ICP_* and the translation matrix *t_ICP_* are calculated.
(10)$$W=Cov(p,q)=\frac{1}{n}\sum_{i=1}^{n}p_{i}q_{i}{}^{T}$$
(11)$$W=U\Sigma V^{T}$$
(12)$$R_{ICP}=VU^{T}$$
(13)$$t_{ICP}=q-R_{ICP}p$$Threshold judging: As shown in Equation (14), the random sample point set *p* is transformed to the correspondence point set *q*. Here, *p*′ constitutes a transformed random sample point set. Next, the error function (*E*) is calculated. The error function constitutes the sum of the Euclidean distances between the transformed point cloud data set *p*′ and the correspondence point set *q*; it is calculated using Equation (15). Here, *p_i_*′ constitutes the *i*th transformed random sample point.



(14)
$$p^{\prime }=R_{ICP}p+t_{ICP}$$





(15)
$$E=\sum_{i=1}^{n}\left\|q_{i}-p_{i}{}^{\prime }\right\|$$



Steps 1–3 are repeatedly executed until the error function is minimized. When the error function becomes less than the error threshold (*ε_ICP_*), the algorithm is terminated. When the algorithm is finished, the entire local point cloud (*P_RANSAC_*) is transformed to the global coordinate system (*P_Global_*), as shown in Equation (16).
(16)$$P_{Global}=R_{ICP}P_{RANSAC}+t_{ICP}$$

### 2.5. Estimation of the Object’s Precise Location in the Global Coordinate System

The last process involves measurement of the object’s 3D position coordinates. The LiDAR sensor trajectory obtained during generation of the local point cloud in Section 2.1 is used in this process. The LiDAR sensor trajectory (*L*) can be expressed in three-dimensional *X*, *Y*, *Z* coordinates for time *τ*, as shown in Equation (17). The interval and range of time *τ* depend on the scanning frequency and LiDAR operating time. The LiDAR sensor trajectory is transformed into the global coordinate system using the rotation matrix and translation matrix calculated in Section 2.3 and Section 2.4. Equation (18) presents the formula that converts the LiDAR sensor trajectory into the global coordinate system. Here, *L_Global_* denotes the final LiDAR sensor trajectory of the global coordinate system.
(17)$$L_{\tau }=\left(X_{\tau },\;Y_{\tau },\;Z_{\tau }\right)\;$$
(18)$$L_{Global}=R_{ICP}(R_{RANSAC}L+t_{RANSAC})+t_{ICP}$$

## 3. Field Experiment

### 3.1. Study Area

The study area comprised the Gwan-in underground mine (38°07′00″ N, 127°13′24″ E), which is located in Gwanin-myeon, Pocheon-si, Gyeonggi-do, Korea. Figure 7 shows an aerial photograph of the study area. The underground mine produces 280,000 tons of iron ore annually via sublevel stoping. The slope of the rampway is approximately 6° to 8°; the width and height of the tunnel are approximately 4.5 m. Iron ore is produced in the 3rd, 5th, 6th, and 7th level drifts. 

### 3.2. Global and Local Point Cloud Datasets

Global and local point clouds of the study area were generated using mobile LiDAR. The mobile LiDAR used in the experiment was GeoSLAM’s ZEB-REVO, A.M. Autonomy’s MAPTORCH, and Apple’s iPad Pro. Three LiDAR systems were used in the field experiment for evaluation of the registration accuracy between point clouds formed from different LiDAR products. The iPad Pro LiDAR was used for the experiment because of its suitability for use in underground mining sites (low cost, light weight, and ease of carrying) in comparison to the other commercialized LiDAR systems. Furthermore, the iPad Pro can be used for data collection, data transmission and reception, and communication between workers in underground mines. Figure 8 shows the appearance of each device and the scene of LiDAR scanning in the study area.

Table 1 presents the specifications of each system. The ZEB-REVO is equipped with Hokuyo’s URM-30LX-F product, which transmits laser signals up to 30 m through a horizontal range of 270° and rotates 360° about the horizontal direction [44]. The MAPTORCH is equipped with Velodyne’s PUCK-LITE 3D LiDAR; it can transmit 16 channels of laser signals through a horizontal 360° area up to a range of 100 m [45]. Both systems form continuous 3D point cloud data using the scan matching algorithm developed by the manufacturer. The iPad Pro uses the flash LiDAR scanning method, in which the photodetector measures the time-of-flight for each pixel in a 2D array field of view [46]. A vertical cavity surface emitting laser sensor with a 10-µm pixel size and 30-K resolution emits the laser; single photon avalanche diodes measure the corresponding distance [47].

A global point cloud was developed for the 6th level drift using the ZEB-REVO. Figure 9 shows the global point cloud of this drift. The 6th level drift is divided into zones A and B according to the ore body distribution; the average elevation of the bottom surface is 65 m above mean sea level. The total number of points in the global point cloud was 9.2 × 10^7^, and the average surface density was 3516 pts/m^2^. The global point cloud was converted into the Korea 2000 Korea Central Belt 2010 TM coordinate system using 10 ground control points of the 6th level drift.

Three point cloud maps were formed for zone B of the 6th level drift using the ZEB-REVO, MAPTORCH, and iPad Pro. The total numbers of points were 1.9 × 10^7^, 9.7 × 10^7^, and 4.5 × 10^7^, respectively; the mean surface density values were 2073, 6412, and 4662 pts/m^2^, respectively. MAPTORCH has a longer scanning range and higher data acquisition rate compared to the other systems; thus, it had the highest number of points in the point cloud map and the highest mean surface density.

Six local point cloud datasets with a length of 20 m were extracted from the point cloud maps in consideration of the drift shape, geometric features, and presence or absence of structures. Figure 10, Figure 11 and Figure 12 illustrate the point cloud maps formed using ZEB-REVO, MAPTORCH, and iPad Pro, along with the local point cloud datasets extracted from each map. The local point cloud datasets include sections with irregular ceiling shapes (section a), curved tunnels (sections b and c), intersections (section d), regular shapes (section e), and railing structures near the catchment area (section f). In Figure 11, there are noise data in the local point cloud datasets at the bottom because of an error in the mapping algorithm. Table 2 presents the amount of point cloud data for each section. 

### 3.3. Results of Field Experiments

By setting the octree level to 10, the resolution of the global point cloud and local point cloud datasets was reduced. The number of the down-sampled global point cloud was approximately 7.0 × 10^5^, and the mean surface density was 15.37 pts/m^2^. Figure 13a shows the amount of data of the down-sampled local point cloud datasets. Table 3 presents parameters utilized for FPFH descriptor calculation and RANSAC and ICP registration. A desktop PC with the Intel(R) Core(TM) i9-9900KF CPU (3.60 GHz) and 16 GB RAM running Windows 10 was used to run the algorithm.

Figure 13 illustrates the amount of the local point cloud datasets and the computation time required to calculate the FPFH descriptor for each section. The computation time of the FPFH descriptor increased as the amount of the point cloud increased. Figure 14a shows the average FPFH descriptor calculated for the global point cloud down-sampled to a voxel size of 0.5 m. Figure 14b–d show the average FPFH descriptor of the local point cloud datasets (sections a–f) obtained from three mobile LiDAR systems. The average FPFH descriptors for each section showed an almost identical histogram form, irrespective of the type of LiDAR. Moreover, they showed a form similar to the form of the average FPFH descriptor of the global point cloud.

Figure 15 is the result of matching the global point cloud and local point cloud datasets through the RANSAC and ICP registration algorithms. All local point cloud datasets are exactly aligned to sections a–f in zone B of the 6th level drift. Irrespective of the type of mobile LiDAR system, it was possible to accurately identify the match area between the global point cloud and the local point cloud datasets.

Figure 16 illustrates the computation time and accuracy of RANSAC and ICP registration. An average interval of approximately 3.71 s was needed for RANSAC registration in the local point cloud datasets, except for section a generated by MAPTORCH. The high computation time of approximately 10 s in section a was caused by the low precision of the local point cloud, as shown in Figure 11b; many iterations were required for algorithm convergence.

To evaluate the accuracy of RANSAC and ICP registration, the inlier root mean square error (inlier RMSE, *I_RMSE_*) was calculated, as in Equation (19). Here, *C* is the set of all inlier correspondences {*p*, *q*}, *N_C_* is the number of all inlier correspondences, and *d_p,q_* is the Euclidean distance between inlier correspondences.
(19)$$I_{RMSE}=\sqrt{\frac{1}{N_{C}}\sum_{(p,q)\in C}^{N_{C}}d_{p,q}{}^{2}}$$

In the local point cloud datasets of ZEB-REVO and iPad Pro, the average *I_RMSE_* values of RANSAC registration were calculated as approximately 0.19 m and 0.2 m, respectively (Figure 16b). In contrast, the average *I_RMSE_* of the local point cloud datasets of MAPTORCH was high (approximately 0.36 m). The noise points present in sections b, c, d, and f deteriorated the RANSAC matching accuracy.

ICP registration required an average computation times of approximately 0.5 s (ZEB-REVO), 1.17 s (MAPTORCH), and 0.73 s (iPad Pro), respectively (Figure 16c). Compared to RANSAC registration, ICP registration required a short computation time because the transformation matrix is calculated by consideration of only the distance error between random correspondences. In addition, the computation time of ICP registration was proportional to the amount of the local point cloud dataset. In the local point cloud datasets of ZEB-REVO, MAPTORCH, and iPad Pro, the average *I_RMSE_* values of ICP registration were calculated as approximately 0.12 m, 0.12 m, and 0.14 m, respectively (Figure 16d). The low *I_RMSE_* values confirmed that precise matching between global point cloud and local point cloud datasets was possible through ICP registration. In addition, MAPTORCH’s local point cloud datasets including noise points could be matched with the global point cloud with high accuracy without additional noise filtering.

Using the coordinate transformation matrix obtained via RANSAC and ICP registration, LiDAR sensor trajectory data for each section were transformed into the global coordinate system. Figure 17 shows the LiDAR sensor trajectories of sections c and e among the local point cloud datasets of ZEB-REVO. The global coordinate changes could be confirmed according to the scanning time of the LiDAR. Through RANSAC and ICP registration, global localization could be achieved by recognition of the user’s location coordinates within the entire underground mining area. In addition, the position change could be observed according to travel time.

## 4. Discussion

### 4.1. Global Localization Considering the Variation in Local Point Cloud Resolution

The sensitivity of RANSAC and ICP registration to the variation in local point cloud resolution was analyzed. The local point cloud datasets (sections a–f) of ZEB-REVO were down-sampled by setting the octree level to values of 6, 8, or 12. The global point cloud was down-sampled by setting the octree level to 10. The drift length of the local point cloud datasets was set to 20 m. The parameters used for the algorithms were identical to the parameters in Table 3.

Figure 18 shows the result of matching the down-sampled local point cloud datasets to the global point cloud via RANSAC and ICP registration by setting the octree level to 6, 8, and 12, respectively. Global localization was possible, regardless of the local point cloud resolution. Figure 19 expresses the computation time and *I_RMSE_* values of RANSAC and ICP registration. As the octree level increased, the point resolution of the local point cloud datasets increased; the computation time of the FPFH descriptor calculation also proportionally increased (Figure 19a,b). The computation time and *I_RMSE_* of RANSAC registration were almost constant for each section, regardless of point resolution (Figure 19c). While the computation time of ICP registration increased as the point resolution increased, the average *I_RMSE_* was constant (approximately 0.12 m) (Figure 19d). Therefore, global localization in the study area was possible with the lowest computation time and highest matching accuracy when down-sampling the local point cloud datasets of ZEB-REVO by setting the octree level to 6.

### 4.2. Global Localization Considering Variation in Local Point Cloud Length

The sensitivities of RANSAC and ICP registration, according to the variation in local point cloud length, were analyzed. New local point cloud datasets were formed from the point cloud obtained with ZEB-REVO by setting the drift length to 10 m and 30 m, respectively. The octree-level to sample the local point cloud datasets was set to 10. Algorithm parameters were set as shown in Table 3. 

Figure 20 illustrates the results of matching the local point cloud datasets with drift lengths of 10 m and 30 m to the global point cloud. In particular, the global point cloud and the local point cloud were accurately matched even under the condition of the 10 m drift length, which generally lacked characteristics of drift shape. The average *I_RMSE_* values were approximately 0.19 m (RANSAC) and 0.12 m (ICP), regardless of drift length variation. The average total computation times required for FPFH descriptor calculation and RANSAC and ICP registration were approximately 8.5 s, 10.3 s, and 10.4 s when the drift length condition was 10 m, 20 m, and 30 m, respectively. Therefore, in the study area, global localization could be achieved with the least time and high accuracy when forming a local point cloud with a length of ≥10 m.

### 4.3. Comparative Analysis with Global Localization Methods

We analyzed and compared the proposed global localization method with an existing method developed by Ren and Wang [34]. Table 4 represents the differences in global localization techniques between the existing method and our method. The existing method generates an offline point cloud map, referred to as the global point cloud in our method, using the GICP-SLAM [32]. On the other hand, we used the graph SLAM [48] technique. Then, the offline map is converted into a distance-weight map (DWM) to correct the offline map error by calculating the coordinates of the point according to the distance between the point and the current LiDAR center. It finds a match area between the DWM and the current local point cloud to estimate an initial LiDAR pose using spatial descriptors including the scan context [49] features of the current local point cloud, key frame, and pose information. Finally, it predicts and corrects the current LiDAR pose on the DWM in real-time using an Unscented Kalman Filter (UKF) by matching the edge and plane features of the underground roadway.

Comparative analysis revealed two major differences in the global localization techniques of the two methods. First, the existing method has focused on real-time implementation of global localization and pose estimation while continuously matching the DWM with the current local point cloud using the UKF. To reduce the computation complexity, it considered the point cloud roadway only. In contrast, our method has only focused on the recognition and measurement of exact 3D spatial coordinates of the current location in the underground mine using point cloud registrations without further and gradual pose estimations.

Next, it has been mentioned that the initial position has a great influence on the real-time positioning [34]. This is because it follows the UKF of predicting the next pose based on the initial position. For this reason, to estimate the exact initial pose, it was preferred that the scan context, key frame, and pose information were used rather than considering the characteristics of drift shape. In contrast, our method used whole point cloud features including ceilings, roadways, and sidewalls to find the exact 3D coordinates of the current location. It would be advantageous for global localization at the underground mine with a wide area on a single level and multiple levels.

For a more quantitative comparison of the two methods, further field experiments can be performed to compare the performances, processing time, relative position error, etc. Experiments may include various technique comparisons between (1) FPFH + RANSAC and scan context, (2) FPFH and other point cloud descriptors, and (3) the use of only roadways and whole underground spaces. In addition, we are planning to advance our method by reflecting (1) further and continuous pose estimation after initial global localization, (2) real-time global localization in the underground mine sites, and (3) cumulative map error compensation using the DWM.

## 5. Conclusions

In this study, we proposed a 3D global localization method that could be implemented using mobile LiDAR mapping and point cloud registration to recognize the location of objects in an underground mine. A prior global point cloud map was generated for the entire underground mine, and the map frame was converted into a global UTM coordinate system. A local point cloud with a minimum length of 10 m was developed using mobile LiDAR. The global point cloud and local point cloud were down-sampled through an octree-based sub-sampling technique; FPFH descriptors were calculated for point feature extraction. The global point cloud and local point cloud were precisely matched via RANSAC and ICP registration; the transformation matrix was calculated to transform the local point cloud to the global UTM coordinate system. Finally, the LiDAR sensor trajectory was converted into the global UTM coordinate system and the object locations were measured. This application of the proposed method to the Gwan-in underground mine confirmed that global matching of local point cloud datasets is possible, regardless of the type of mobile LiDAR system and geometry of the underground drift. Through the RANSAC and ICP registration, the result was precisely aligned with an average inlier RMSE error of approximately 0.13 m between global point cloud and local point cloud datasets. In addition, by converting the LiDAR sensor trajectory data into a global UTM frame, the spatial location coordinates of an object could be recognized according to travel time.

The proposed method can provide innovative solutions for 3D position recognition in the Gwan-in underground mine. In the study area, managers have adopted only traditional localization methods by using the total station instrument and installing artificial landmarks. Only about ten or more artificial landmarks are installed on the ceiling on a single level. Because they are memorizing all locations of the mine, they are not obligated to adopt advanced localization methods such as installing wireless signal sensors and implementing the LiDAR SLAM. However, they are increasingly seeing the need to configure digital twins and install mine automation systems to improve mine productivity and sustainability. Using cost-effective and novel tools including 3D mobile LiDAR mapping systems and point cloud registration methods, they can immediately and accurately track the 3D spatial coordinates of assets, processes, and environments in the study area.

An analysis of the accuracy and computation time of the 3D global localization method through field experiments confirmed that this method can be utilized effectively in an underground mine site. The proposed 3D global localization method has the following contributions for the mining industry to become automated, intelligent, and digitalized.

It enables improvement of replication precision and analysis/prediction/automation accuracy of the digital twin platform. In particular, accurate analysis and prediction for overall mining operations may be possible through accurate spatial conversion of geological survey (e.g., ore mapping and borehole), production (e.g., blasting patterns and ore volume), and geotechnical (e.g., rock fall and road conditions) data into the digital twin platform.In addition, the operating accuracy of autonomous driving equipment can be increased via sophisticated 3D position recognition in underground mines composed of multiple drift levels.To generate the local point cloud map, low-cost, open-source 3D LiDAR can be used. Recently, various low-cost 3D LiDAR products have been commercialized for self-driving robots. They can build the local point cloud with a minimum length of 10 m and appropriate point resolution. They are also sufficiently small and rigid to attach to a worker’s helmet, equipment bodies, or facilities.Furthermore, there is no requirement for auxiliary sensors (i.e., inertial measurement units) for continuous mapping and pose estimation in an underground mine with high magnetic field intensity.

In the future, we plan to advance the proposed system so that it can be operated in the underground mine in real-time. At the current level, it is insufficient to implement the system in real-time due to the computation complexity of the algorithms, the vastness of point cloud data, and the use of individual 3D mobile LiDAR and processors. The hardware and software would be configured to simultaneously enable LiDAR mapping, point cloud registration, and visualization on one processor. In addition, it would be possible to estimate continuous pose estimation through continuous scan-to-global point cloud map matching after initial global localization. To evaluate the applicability of the 3D global localization method, additional field experiments will be performed on underground mines that have various drift geometries and are of different types (e.g., coal mines, limestone mines, etc.).

## Figures and Tables

**Figure 1 sensors-22-02873-f001:**
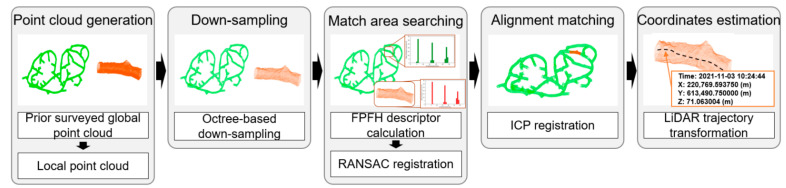
Schematic diagram describing the procedure of 3D global localization.

**Figure 2 sensors-22-02873-f002:**
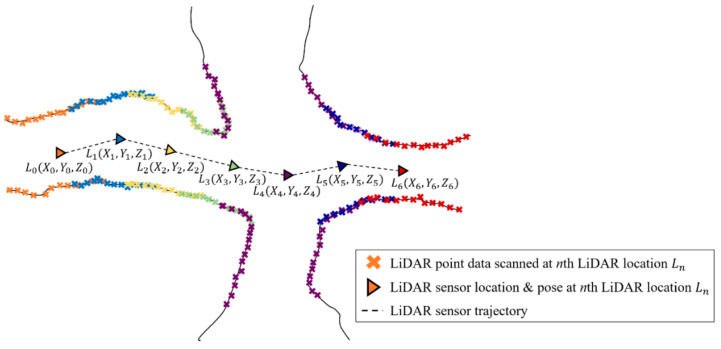
Description of point cloud generation and LiDAR sensor trajectory via scan matching.

**Figure 3 sensors-22-02873-f003:**
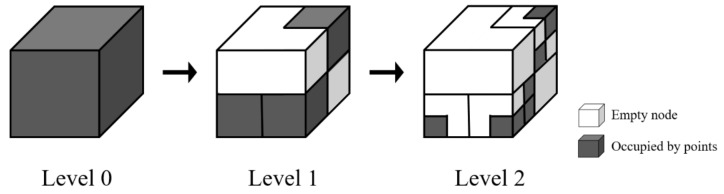
Description of octree-based subdivision.

**Figure 4 sensors-22-02873-f004:**
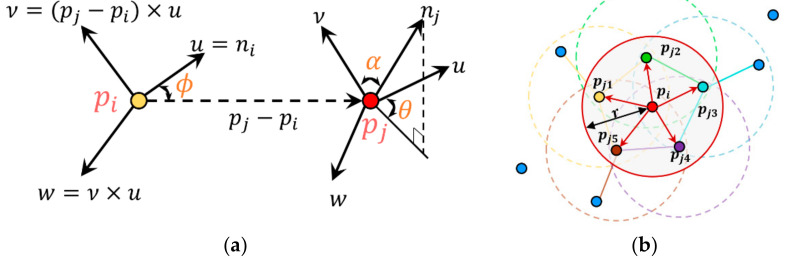
Schematic diagram for FPFH descriptor computation and RANSAC registration: (**a**) a graphical formulation of the Darboux uvw frame and (*α*, *ϕ*, *θ*) angular variation; (**b**) the effect area (red solid line) for computation of the FPFH descriptor of *p_i_* with five nearest neighbors. Colored dotted lines present the effect area of each nearest neighbors.

**Figure 5 sensors-22-02873-f005:**
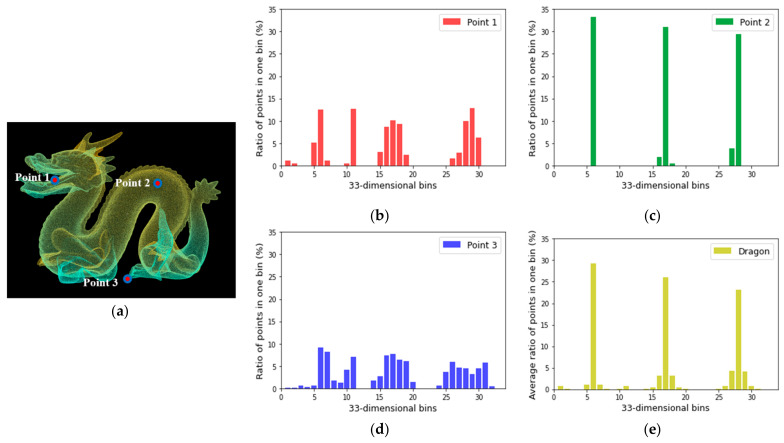
Example of FPFH descriptor: (**a**) dragon point cloud dataset (provided by Stanford University Computer Graphics Laboratory); FPFH descriptor of (**b**) point 1 (mouth), (**c**) point 2 (body), and (**d**) point 3 (foot); and (**e**) average FPFH descriptor of whole dragon point cloud data.

**Figure 6 sensors-22-02873-f006:**
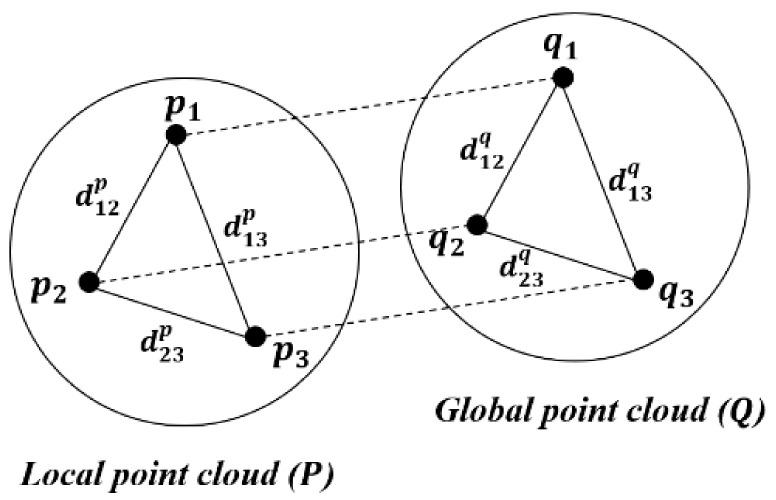
Graphical diagram of relative dissimilarity vector computation when including three corresponding points {(*p*_1_, *q*_1_), (*p*_2_, *q*_2_), (*p*_3_, *q*_3_)}.

**Figure 7 sensors-22-02873-f007:**
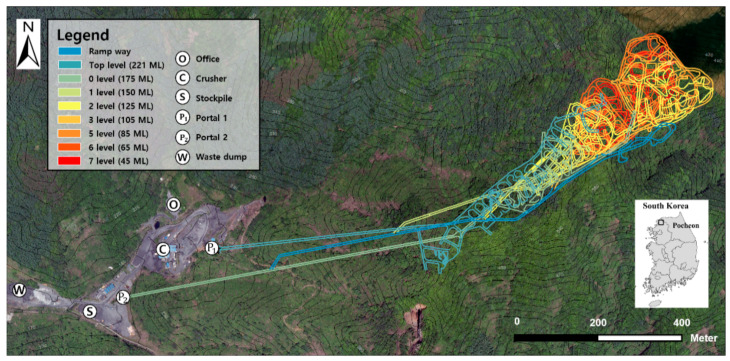
Graphical diagram Aerial view of the study area (Image source: National Geographic Information Institute).

**Figure 8 sensors-22-02873-f008:**
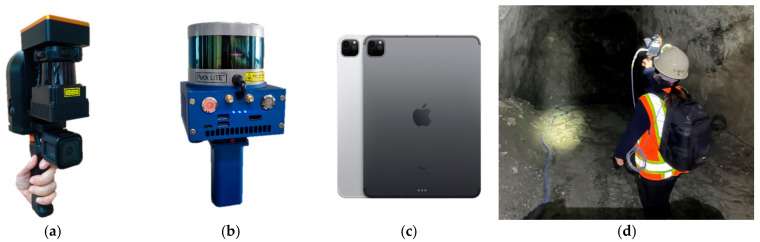
The mobile LiDAR systems used for field experiments in the study area: (**a**) ZEB-REVO (GeoSLAM; London, UK) (**b**) MAPTORCH (A.M. Autonomy; Seoul, Korea); (**c**) iPad Pro (Apple; Santa Clara, CA, USA; image source: https://www.apple.com (accessed on 6 April 2022); (**d**) scene of the mobile LiDAR scanning in the study area.

**Figure 9 sensors-22-02873-f009:**
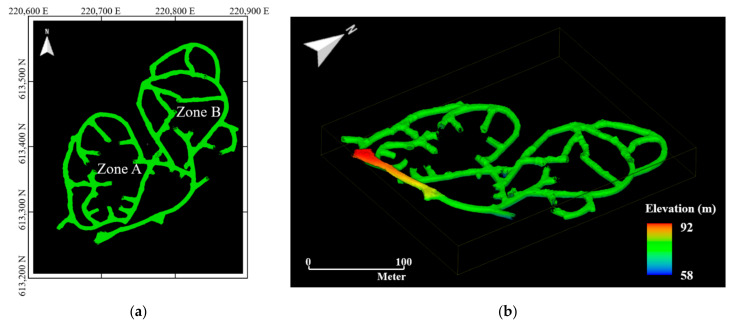
Global point cloud map of 6th level drift generated using ZEB-REVO: (**a**) Plan view of 6th level drift. Reference grid is in meters. The 6th level drift is divided into two zones (A and B) due to the ore body distribution. Local point cloud datasets were generated for the zone B. The origin of the local transverse Mercator coordinate system is 38°00′00″ N, 127°00′00″ E (map datum: GRS 80); (**b**) 3D top view of 6th level drift.

**Figure 10 sensors-22-02873-f010:**
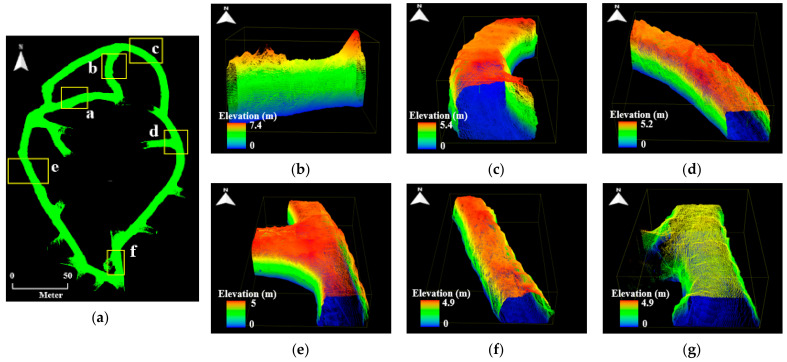
Local point cloud datasets generated using ZEB-REVO: (**a**) point cloud map of zone B of 6th level drift; (**b**) section a; (**c**) section b; (**d**) section c; (**e**) section d; (**f**) section e; (**g**) section f.

**Figure 11 sensors-22-02873-f011:**
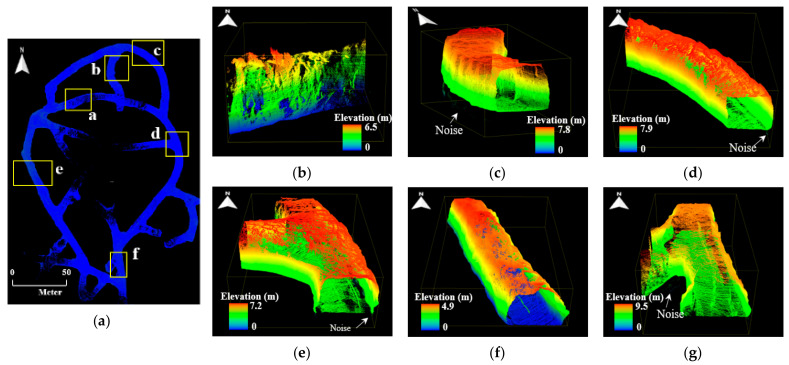
Local point cloud datasets generated using MAPTORCH: (**a**) point cloud map of zone B of 6th level drift; (**b**) section a; (**c**) section b; (**d**) section c; (**e**) section d; (**f**) section e; (**g**) section f.

**Figure 12 sensors-22-02873-f012:**
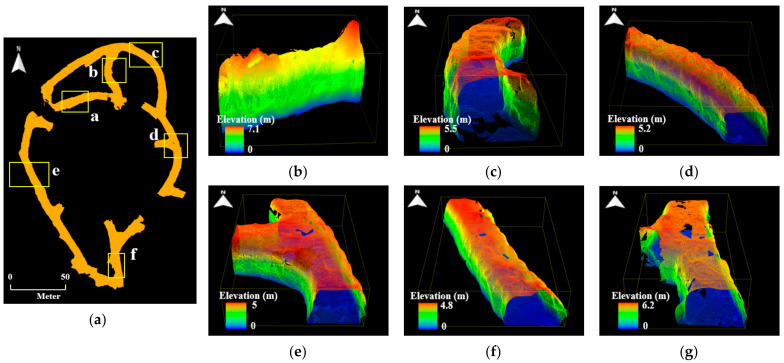
Local point cloud datasets generated using iPad Pro: (**a**) point cloud map of zone B of 6th level drift; (**b**) section a; (**c**) section b; (**d**) section c; (**e**) section d; (**f**) section e; (**g**) section f.

**Figure 13 sensors-22-02873-f013:**
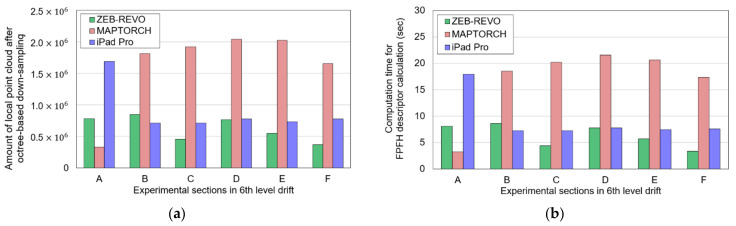
Results of octree-based down-sampling and FPFH descriptor calculation: (**a**) amount of local point cloud after down-sampling; (**b**) computation time of FPFH descriptor calculation (s).

**Figure 14 sensors-22-02873-f014:**
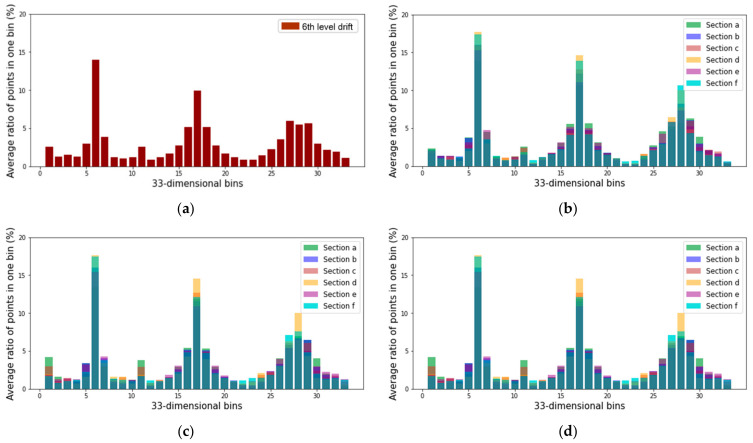
FPFH descriptor of the global point cloud and local point cloud datasets: (**a**) global point cloud; (**b**) local point cloud dataset generated by ZEB-REVO; (**c**) local dataset generated by MAPTORCH; and (**d**) local dataset generated by iPad Pro.

**Figure 15 sensors-22-02873-f015:**
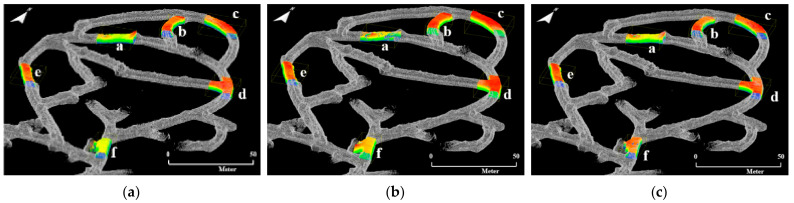
Experimental results of RANSAC and ICP registration between the global point cloud and local point cloud datasets of section a–f: (**a**) ZEB-REVO; (**b**) MAPTORCH; (**c**) iPad Pro.

**Figure 16 sensors-22-02873-f016:**
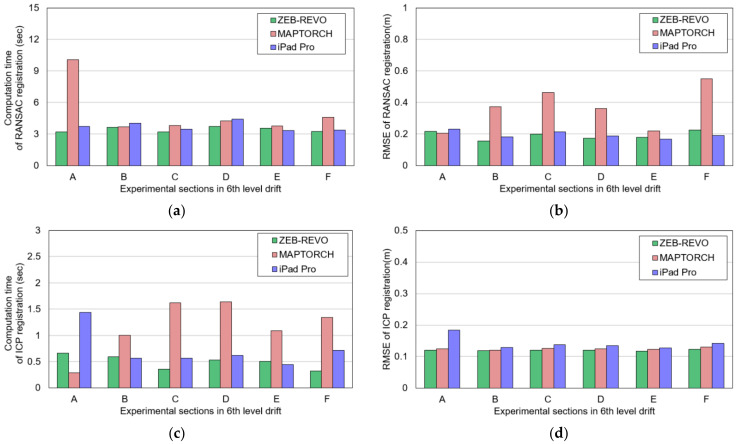
Experimental results of RANSAC and ICP registration: (**a**) computation time of RANSAC registration; (**b**) inlier RMSE of RANSAC registration; (**c**) computation time of ICP registration; (**d**) inlier RMSE of ICP registration.

**Figure 17 sensors-22-02873-f017:**
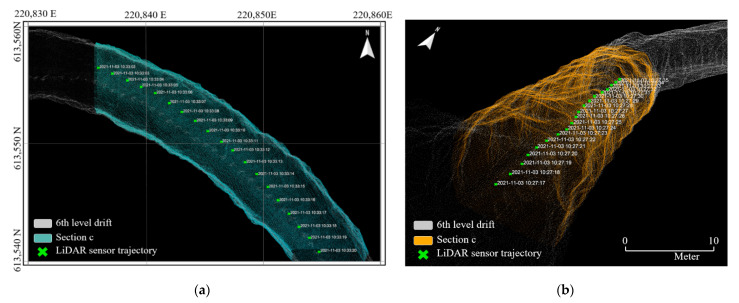
Experimental results of estimating user’s location coordinates on the global coordinate system according to the travel time: (**a**) top view of section c; (**b**) 3D view of section e.

**Figure 18 sensors-22-02873-f018:**
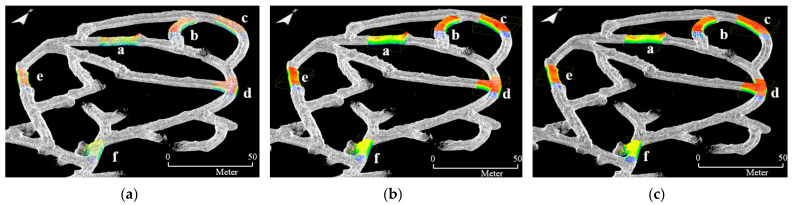
Experimental results of RANSAC and ICP registration considering octree level variation of the local point cloud datasets of section a–f: (**a**) octree level 6; (**b**) octree level 8; (**c**) octree level 12.

**Figure 19 sensors-22-02873-f019:**
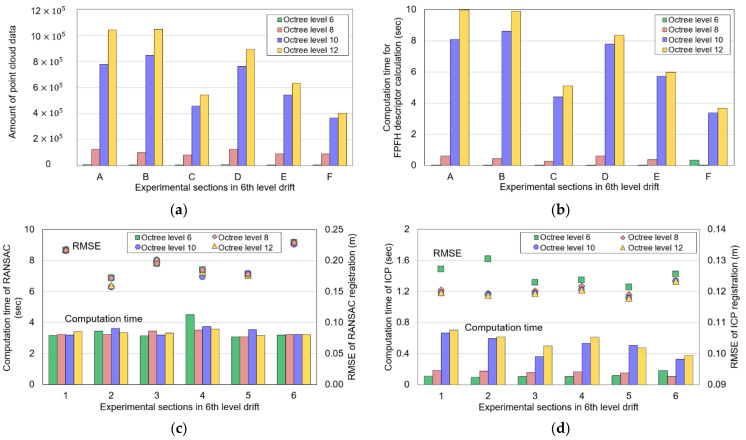
Experimental results of FPFH descriptor calculation, RANSAC registration, and ICP registration considering variation in the local point cloud resolution: (**a**) amount of the local point cloud; (**b**) computation time of FPFH descriptor calculation; (**c**) computation time and inlier RMSE of RANSAC registration; (**d**) computation time and inlier RMSE of ICP registration.

**Figure 20 sensors-22-02873-f020:**
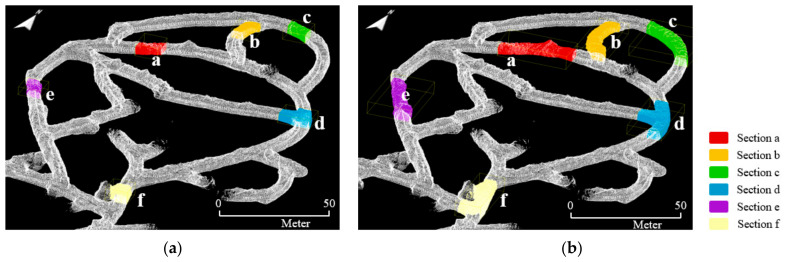
Experimental results of RANSAC and ICP registration considering variation in drift length of the local point cloud datasets: (**a**) 10 m and (**b**) 30 m lengths of local point cloud datasets.

**Table 1 sensors-22-02873-t001:** Specifications of mobile LiDAR systems used for field experiments.

Specifications	ZEB-REVO	MAPTORCH	iPad Pro 11
Manufacturer	GeoSLAM	A.M. Autonomy	Apple
Country of manufacturer	London, UK	Seoul, Korea	Santa Clara, CA, USA
LiDAR product	Hokuyo UTM-30LX-F	Velodyne PUCK-LITE	dToF sensors
Scanning method	Motorized optomechanical	Motorized optomechanical	Flash LiDAR
Maximum scanning range	30 m	100 m	5 m
Data acquisition rate	43,200 points/s	0.3 million points/s	10 µm pitch pixel
Resolution	0.625° horizon, 1.8° vertical	0.1° horizon, 2° vertical	0.03 MP resolution
Field of view (FOV)	270° horizon × 360° vertical	360° horizon × 30° vertical	122° horizontal
Rotation rate	0.5 Hz	5–20 Hz	-
Laser wavelength	905 nm	905 nm	-
Additional sensors	Microstrain 3DM-GX2	Microstrain 3DM-GX5	accelerometer, gyro sensor
Accuracy	±3 cm	±3 cm	-

**Table 2 sensors-22-02873-t002:** Amount of point cloud data according to the local point cloud sections generated by ZEB-REVO, MAPTORCH, and iPad Pro.

Type of System	The Amount of Point Cloud Data (unit: 1.0 × 10^6^)					
	Section a	Section b	Section c	Section d	Section e	Section f
ZEB-REVO	1.05	1.05	0.54	0.89	0.63	0.40
MAPTORCH	0.38	2.81	3.95	2.83	3.63	2.21
iPad pro	6.02	1.06	0.92	0.91	1.09	1.16

**Table 3 sensors-22-02873-t003:** Parameters set for implementing FPFH descriptor calculation and RANSAC and ICP registration.

Process		Parameters	Value
FPFH descriptor	Down-sampling	Voxel size	0.5 m
	Surface normal estimation	Maximum number of k-nearest neighbors	1000
		Sphere radius (*r*)	1 m
	FPFH feature calculation	Maximum number of k-nearest neighbors	1000
		Sphere radius (*r*)	2.5 m
RANSAC	Pruning	Relative dissimilarity vector threshold (*ε_vector_*)	0.9
		Distance threshold (*ε_distance_*)	2.5 m
		The number of corresponding point sets to fit RANSAC	4
	Convergence criteria	Maximum number of iterations	4.0 × 10^8^
		Maximum number of validations	500
ICP	Pruning	Maximum correspondence points-pair distance	0.5 m
	Convergence criteria	Maximum number of iterations	30
		Relative RMSE	1.0 × 10^−6^

**Table 4 sensors-22-02873-t004:** Comparative analysis of global localization techniques used in our method and the existing method proposed by Ren and Wang [34].

Techniques		Ren and Wang [34]	Ours
Mapping	Global point cloud generation	GICP-SLAM [32], DWM	Graph SLAM [48]
Initial pose estimation	Descriptor	Scan Context [49], Key frame and pose information	FPFH
	Estimator	Point cloud matching method	RANSAC
Localization	Descriptor	Edge and plane features of roadway	-
	Estimator	Unscented Kalman Filter	ICP
	Real-time or not	O	X

## Data Availability

In this paper, publicly archived datasets were used to show how FPFH descriptor works for point feature extraction and analysis. The dataset is available at the following link: http://graphics.stanford.edu/data/3Dscanrep/ (accessed on 6 April 2022).

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
