# Peer review of "3D Global Localization in the Underground Mine Environment Using Mobile LiDAR Mapping and Point Cloud Registration"

_sensors, 2022, doi:10.3390/s22082873_

Round 1

Reviewer 1 Report

  1. Please note the problem of format. The format of some title of this article is inconsistent, and the serial number of the title is also wrong.
  2. In step 3 of section 2.3.2 of the article, the explanation of step 3 is not very clear, so modification is recommended.
  3. There were errors in the interpretation of the equation (13).
  4. If conditions permit, comparative experiments can be conducted in other types of mines (e. g., coal mines).

Author Response

Thank you for your valuable comments. We revised the original manuscript according to all your comments. Please see the attachment. 

Reviewer 2 Report

It is well known that the global localization process, which estimates the location in the global coordinate system by matching and aligning a pre-built point cloud map and local LiDAR scanning through point feature extraction and point cloud registration, has greatly improved the estimation of 3D map positioning of an unmanned vehicle matching to a predefined point cloud map and local LiDAR scanning on city roads and outdoors.
Thus, the authors, citing that no studies have used LiDAR-based global 3D localization for recognition
location of an object in an underground mine were inspired by the goal to present a 3D global localization method that uses mobile LiDAR mapping and point cloud registration to recognize the location of an object in an underground mine. This method searches for an area of agreement between a prebuilt underground mine point cloud map and a local point cloud through random sampling consensus (RANSAC) and iterative nearest point registration (ICP).
In addition, it evaluates the position of the object in the global coordinate system. They presented the results of field experiments using three mobile LiDAR systems in the Gwang-in underground mine in South Korea.

However, unlike quarries and surrounding spaces, underground workings, along with under-roofing premises, can be successfully equipped with laser and artificial landmarks to indicate the coordinates of special points. It doesn't seem to result in much installation overhead, but it can greatly improve the overall positioning resolution. Therefore, I dare to wonder why not test these known possibilities to overcome the inevitable difficulties in applications of the authors' approach.

Author Response

Thank you for your valuable comments. We also appreciate your meaningful evaluation for our research. We revised the original manuscript according to all your comments. Please see the attachment. 

Reviewer 3 Report

Authors contributions:

The authors have proposed a 3D global localization technic that implements mobile LiDAR mapping and point cloud registration to recognize the locations of objects in an underground mine.

An initial global point cloud map is built for an entire underground mine area using mobile LiDAR.

To generate the local point cloud map, low-cost, open-source 3D LiDAR can be used.

3D global localization method has been proposed. It has some advantages:

  • Enables improvement of replication precision and analysis accuracy of the digital twin platform;
  • Accurate analysis and prediction for overall mining operations may be possible through accurate spatial conversion of geological survey, production and geotechnical data into the digital twin platform;
  • The operating accuracy of autonomous driving equipment can be increased via sophisticated 3D position recognition in underground mines composed of multiple drift levels.

The effectiveness of the proposed method is accessed in real mine environment.

I have some reviewer notes:

Figure 8. It will be good to show the manufacturer and country of origin of the presented equipment.

Figures 13, 16, 19. The sections A, B, C, D, E and F are different. That’s why it is not good to be presented on chart with connected values on the graph. Bar charts are better for their representation. For example, connected points can be used when time sequence is presented on horizontal “x” axis. But for data with different meaning it is not good to connect their points.

Discussion part. You have to compare your results with those from other authors. Minimum 3 papers must be compared.

Conclusion part. It is not clear how your results improve the known solutions in this study area. Also, you have to describe how your work will be continued.

I have some suggestions:

It will be good to make more comparative analyses with papers from other authors. It will improve your contributions.

Author Response

Thank you for your valuable comment. We also appreciate your meaningful evaluation for our research. We revised the original manuscript according to all your comments. Please see the attachment. 
